# Mining Braces of Innovation Linking to Digital Transformation Grounded in TOE Framework

**Fumeng Li** [1,*] 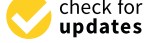**, Jiancheng Long** [1] **and Wu Zhao** [2]

1   Department of Economics & Management, Xidian University, Xi'an 710071, China
2   School of Marxism, Xidian University, Xi'an 710071, China
*   Correspondence: lifumeng28@163.com

**Abstract:** How firms drive innovation in digital transformation remains largely unanswered and this article is an attempt in that direction to deconstruct the digital innovation of small and medium-sized manufacturing enterprises (SMMEs) realizability condition and evolve the body of knowledge. We developed a TOE framework based on digital innovation theory to investigate the impact of the configuration effect of technology, organization and environment regarding the characteristic on a firm's digital innovation. We performed fuzzy-set qualitative comparative analysis (fsQCA) on survey data collected from 141 SMMEs in China to examine configuration paths formed by different conditions. The results reveal that the success of a firm's digital innovation practice is not driven by a single factor, but the result of multiple factors' combined interaction, in which four sets of high digital innovation realization paths could be further summarized as "total factor driven", "technology-environment oriented", "organization-technology oriented", and "organization oriented-environment". These findings make sound theoretical and practical contributions to the usage of the TOE framework in the domain of developing a firm's digital innovation. Bringing the SMMEs' enlightenment is digital innovation, which is integral, systematic engineering, despite technology itself being the primary role of the whole process, more important is the organization's agile strategy and digital positioning, as well as making full use of the advantages of the current environment for companies, thus better promoting the emergence and deepening digital innovation.

**Keywords:** digital innovation; TOE framework; SMMES; fsQCA



## 1. Introduction

Digital technology permeates and enables the rapid development of the industrial economy. It is an axiomatic fact for most organizations that enterprises are forced to join the wave of digital transformation and innovative development [1,2], which has led to fundamental changes in their products, processes, organizations, and business models [3,4]. Although digital transformation has changed the elements of enterprise innovation and in turn provides the infrastructure, key technologies, and application platforms for digital innovation, the emergence of digital solutions has also questioned the interpretive framework of existing innovation theories [5,6]. In view of the continuous expansion of the digital map built by IT-embedded and intelligent products and services, more firms have begun to move from primary transformation to the deep application stage. Considering that digital transformation is a fundamental revolution process that changes entity attributes via digital technology, how to make continuous use of resource advantages after transformation to create a subversive competitive model that breaks vertical industrial silos and creates networks [7] is a subject of common interest in the field of information systems and innovation [8]. Digital innovation requires the co-engagement of heterogeneous agents to innovate by continuously combining and recombining different digital technology components [5,9]. Importantly, digital innovation emphasizes the potential to include the participation of different stakeholders. In other words, digital innovation could be defined as the process

of social technology innovation or entrepreneurship in which different stakeholders apply digital technology to the current environmental and institutional context in different contexts [10]. For example, community building relies on digital technology innovation to connect and communicate with a wide range of beneficiaries, becoming an important resource for enterprises to create sustainable innovation [11].

Undeniably, although the engagement of enterprises in digital innovation highlights attractive opportunities, it also implies that it is an unknown challenge and radical strategy for survival [12], as well as innovation results and processes need to be relentlessly deconstructed and reconstructed [5,13]. For making full use of the technology and resources in the iterative development, enterprises not merely need to constantly update their thinking and understanding of digital tasks, but as requested to create conditions of open and generativity for digital innovation [14]. The current competitive landscape, in fact, is being disrupted and subverted by new digital technologies, accompanied by knowledge heterogeneity and skill diversity contained in a technology, making its complexity a common feature faced by many enterprises in their innovation practices [15]. Obviously, the interlaced characteristics of dynamic, self-referential, malleable, and editable enable innovative actions for sustainable development [16]. Digital technology, with aspects of data homogenization and reprogrammable functionality [9,16], are available for different organizations and individuals to achieve different goals using the same digital technology, that is, technology affordance [17]. The enhancement of the breadth and depth of the knowledge search provides support for technological innovation after breaking the flow barrier of boundary heterogeneous resources [18].

Diversified consumer demands, sharp shortening of the product innovation cycle [19], and highly discrete knowledge [20] in the digital era urge enterprises to enhance the breadth and speed of resource acquisition, integration, and allocation. As proof of this, organizations reallocate resources in a faster and easier way by continuously updating their agile capabilities to search, explore, acquire, assimilate, and apply relevant knowledge, which leads an organization to efficiently and effectively redeploy/redirect its resources to value creating and value protecting (and capturing) higher-yield activities as internal and external circumstances warrant [21]. In the process of external knowledge mining, screening, acquisition, and utilization, organizational learning is considered to be a decisive factor for restructuring organizational structure, driving strategic renewal, and achieving sustainable competitive advantage [22].Considering the impact of digitalization on the conventional cognitive model of enterprises, for some traditional industries, organizational inertia due to the inherent practices formed over the years would refuse the potential dividends provided to enterprises by digitalization [23]. Therefore, organizational unlearning calls for special attention, and will replace the dominant logic shaped in the start-up period, reasonably in turn building a value model more compatible with digitalization [24]. What is obvious is that the stronger unlearning ability conduces to make appropriate and positive strategies in resource restructuring for enterprises, together with sustained economic and social benefits.

The development of the digital environment is constantly updated and unpredictable. Specifically, volatility may cause the loss of the competitive advantages that organizations rely on to survive and lead them into business crisis, accompanied by bringing opportunities for the application of new technologies, products and services, thus forming new competitive advantages [25,26], which is also the original driving force for organizations to stimulate dynamic capabilities and transform into innovative advantages [27]. Digital innovation in the fuzzy boundary, more importantly, cannot be separated from the continuous identification, absorption, transformation, and integration of external resources, while the abundance and difficulty of obtaining the key resources required by enterprises in the business environment significantly affect the value creation and commercialization activities of enterprises [28]. Thus, it can be seen that firms have to deal with the challenging identification, utilization, rearrangement of their resources, to create and develop new digital offerings. From this viewpoint, embracing the technological, organizational,

and environmental characteristics of digitalization is closely linked to the generation of relevant innovation and creative outcomes. Although quantitative analysis of extracting single variable and traditional case studies provide fertile ground for digital innovation management, with the focus on configuration problems and set relationships, qualitative comparative analysis (QCA) is a promising approach to study new mechanisms of interaction among different influence routes [29]. Some scholars extend the technology–organization–environment (TOE) framework to study the correlation between business processes, digital platform building, and innovation [30,31], contributing to broaden understanding of the micro foundations of digital innovation. Since digital innovation requests that firms both create new technical combination and renew organizing capabilities [5], preparedness for such capabilities takes on increasing relevance. For this reason, taking into particular account the new technology adoption view, the technology–organization–environment framework appears to be a powerful lens through which to investigate how organizations make strategic reserves based on digitalization and, consequently, constitutes as an important source of innovation, which becomes the value co-creation between companies and other actors as well as realizing a new form of social adventure participation.

In sum, this research starts from the question of "What are the key elements supporting digital innovation from the perspective of characteristics and conditions and what are their linkage effects?" to narrow the knowledge gap. Digital innovation involving multiple subjects is a dynamic and iterative process based on technology, internal and external organization, as is evident in the literature, which further determines the research objectives—employing a comprehensive framework to investigate which technology–organization–environment related sub-conditions constitute the digital innovation paths of different configurations. Despite this, and unpredictably, to the best of our knowledge, the conditions with the TOE framework of firms realizing digital innovation has received limited attention from academics, or, in other words, less comprehensive and specific research [6,17,32]. In addition, studies examining how organizations build multiple factors for digital innovation are scant, especially in the context of SMMEs. Therefore, an expanded discussion of digital innovation contributes to a scientific knowledge base on relevant and recent topics, which is of interest to both academics and practitioners [33]. Furthermore, it is also consistent with the view and acknowledges advocated by the strategic management field and innovation scholars—producing innovation is not merely a simple subject to traditional enterprise resources support and dynamic capability, more than that, by force of nourish and blessing [5,6,21]. The key practical value we anticipate is to better understand the impact of digitization in the context of SMMEs, providing insights on driving digital innovation based on TOE models via qualitative research. In the long term, enriching the literature on digital innovation management by investigating how companies are re-thinking their innovation models and adopting digital solutions, these solutions, while creating new products and services for customers, would constitute the core competitiveness to sustain sustainable development in the process of deep application of digitalization in companies.

The article is structured as follows. After the introduction, it reviews literature on the topic of digital innovation, as well as its realizable condition within the TOE framework, examining the main conditions—technological affordance and complexity, organization-al agility and unlearning, environmental dynamics and munificence—strongly associated with the firms' digital innovation. Then, a discussion of the methodology follows, before the article moves on to the findings, describing the research design and measurement used for the empirical analysis, together with the data collection from a sample of 141 firms in China. Subsequently, the results of the fsQCA are presented and discussed, highlighting the four configuration paths for high digital innovation and one non-high digital innovation. In the final sections, theoretical and managerial implications, as well as some limitations, are presented and discussed.

## 2. Literature Review and Theoretical Framework

Digital innovation theory is the core of the theoretical framework of this paper. Scholars in the field of information management and innovation have given some conceptual explanations to it [3,5,6,34], which can lead to new market offerings, production processes, organizational patterns, and business processes or models that result from the use of digital technology. Indeed, given that digital technology fundamentally changed consumer expectations and behavior, together with upending incumbents' competitive landscape, digital innovation essentially could be viewed as a disruptive innovation for companies to cater to the market and maintain competitiveness [7], which blurs the boundaries of participants and information exchange (convergence), forming a dynamic capacity to continuously improve and change to produce unprompted change driven by large, varied, and uncoordinated audiences (generativity) [17]. With the recent widespread attention on the initiation and implementation of digital innovation in organizations, several studies have fully affirmed the responsibility of digital technologies in promoting innovation convergence and generativity, as well as the contribution of innovation to outcomes [35], As evidence, IT artifacts and designed infrastructure are the original conditions used to create new combinations of physical and digital products [33]. On the contrary, the significant changes within the organization after the digital transformation of enterprises make it possible to develop new products, services, and processes [36], in which resources, information technology, cognition, partners, organizational culture, and strategy could be used as preparatory factors for the smooth implementation of digital innovation [37]. Similarly, some studies pay more attention to the joint role of the internal and external environment in the digital innovation process, in which it seems that the real key is to obtain the available technical knowledge from the resources in the environment and continuously apply, learn and share it to achieve the iteration of innovation [38]. Despite being insightful, existing pieces of evidence have widely regarded existing technology of the organization and environment as having a significant impact on digital innovation; studies that address whether firms are co-driven by different elements that drive the innovation process are limited. With reference to the aforementioned points, investigating the combined impact and interactivity of digital technologies, organizational patterns, and environmental characteristics may jointly define an interesting theoretical framework for digital innovation.

A framework that would be beneficial in facilitating the understanding of digital transformation and innovation success is the technology–organization–environment (TOE), which classifies three areas that affect the process by which organizations implement or adopt technology. TOE has advanced to become a valuable theoretical lens for understanding new technology adoption for applicability in an organization [39] and a widespread theoretical perspective related to performance and innovation [40–42]. It is worth noting that the TOE framework does not identify specific factors, as these should be determined according to the research setting and research questions. Considering that the theoretical difficulty in QCA configuration analysis is indeed how to identify configuration conditions, including two construction methods of induction and deduction [43], we choose the exploratory induction method that relies on past literature and empirical knowledge, together with the deductive method based on a certain theoretical framework (TOE model).

We attempt to understand the goals of specific factors by analyzing the characteristic indicators of digital technology and the organizational environment. While the technological characteristic describes the digital technology-related factors, including technological affordance and complexity relevant to the institution [44], the organizational context explains the descriptive measures of the implicit indications involving agility and unlearning [45], and the environmental context refers to external digital factors having significant influence on a firm's innovation beyond its control such as dynamics and inclusivity [46]. Thus, this review has identified the TOE framework to be mapped with the constructs of the digital innovation to establish the likely success dimensions for innovation (Figure 1).

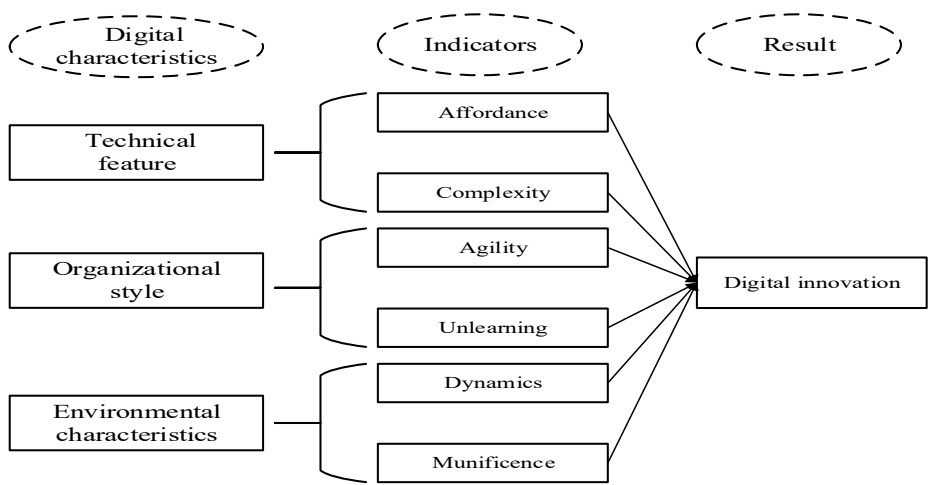

**Figure 1.** Research model.

## 2.1. Technological Affordance and Complexity

As a frontier technology research topic in the field of digital innovation, digital technology affordability refers to the potential behavior possibilities provided by digital technology relative to specific entities [17]. What loomed large for theorists and entrepreneurs was how and why the same digital artifact, digital platform, or digital infrastructure can lead to different innovations in different usage contexts [5]. The redesign process of value creation, delivery, and capture in the entrepreneurial ecosystem is understood from the perspective of availability [47]. The process of homogeneity and heterogeneity of digital technology affordance, by its very nature, gives rise to different paths of digital innovation realization [48]. Data and information represented by binary digits (homogeneity) facilitate enterprises to communicate and collaborate with other organizations under the same standard. Similarly, the programming that facilitates digital devices (reprogrammability) enables different enterprises to achieve a combination of digital technologies. The perspective of availability may help us understand the internal logic of technological characteristics and innovation paths of enterprises.

Technological complexity is a problem facing many industries at present, which has become a powerful weapon to solve problems such as organizational decision-making and technological innovation in complex environments [49]. Digital technology presents complex features, such as openness, distribution, dynamic, self-referential, malleable, and editable [6,16,21], which provide a variety of options for enterprises to realize innovation paths [15]. Judging from an objective perspective, digital technology itself contains components, links, and the association with external resources, where the boundary is fuzzy, the interaction is dynamic, and the contained knowledge is diversified, which is conducive to the self-growth of innovation [17]. The fact is that some digital products can carry out real-time iterative innovation according to user feedback and various problems in the operation process strongly prove this point [9]. If measured from the perspective of technology understanding and application, technical complexity means overloaded influx of new technologies and insufficient knowledge of technologies, which is undoubtedly a double-edged sword for enterprises [50,51]. In general, there is room for theoretical discussion to take technology affordance and complexity as the secondary characteristic indicators to measure "technology". The affordance satisfies the unique value of digital technology application and promotion, while the complexity is widely regarded as the reserve condition for companies to introduce new technologies and implement new strategies. The nonlinear interaction among elements inevitably puts forward higher requirements for the innovation process and results.

## 2.2. Organizational Agility and Unlearning

The embedding of digital technology not only leads to a significant increase in the complexity of knowledge based on product and service innovation, but also puts forward shorter requirements for the product generation cycle [20,52]. Organizational agility is always regarded as a core capability for organizations to quickly respond to changes in the external environment [52], which can be understood as a perceptual and responsive behavioral capability that allows enterprises to redesign existing processes and quickly create new ones. Agility, except for responsiveness, also supports the process of continuously adapting to proactively identify emerging business opportunities in a changing context relative to competitors [53]. The structure of agile organizations requires a flexible market at the front end and a stable middle and back end, accompanied by reference information systems that fundamentally change the management logic of traditional business processes. More importantly, it stimulates the creation of new resources via different resource restructuring and guides the organization into the iterative cycle of digital innovation in the agile development process of constantly exploiting innovation opportunities [54].

Innovation is path-dependent, inevitably, enterprises seek innovation by following past success paths and paradigms [55]. Automatically, the fuzzy treatment of the boundary between subjects and links in digital innovation, accompanied by the continuous iterative process of scene integration [6], may make the outdated management paradigm of the enterprise constrain the vision and development strategy, resulting in the hindrance of the enterprise's sustainable innovation. Previous research has shown that traditional firms that are better able to unlearning emerge strong, with the ability to survive, adapt, and renew in the digital age [56]. This learning model represents the abandonment of past cognition, dominant logic, conventions, etc., to make room for subsequent learning [24], which is significantly applicable to the process of digital innovation iteration and improvement. It is necessary to admit that organizational learning plays a key role in promoting the continuous iteration of innovation. Rooted in generativity of digital innovation, continuous updating of situational cognition and knowledge exploration is an important basis for stimulating innovation update iteration [57], thus embracing digitalization cognitively and behaviorally by changing information sharing mechanisms and workflows. It is obvious that agility and unlearning as the key elements of organizational competence are in line with the new dynamic competence theory under digitalization, which is significantly associated with technological competence, uncertain environment, and innovation performance [58,59].

## 2.3. Environmental Dynamics and Munificence

The business environment driven by digital technology generally presents the characteristics of variability, complexity, uncertainty, and ambiguity [60]. Supporters of dynamic capabilities theory emphasize that the change of competitive advantage created by organizational capabilities depends on environmental dynamics [21]. Rapid changes in consumer demand, industry competition, policy support, or crackdown would bring more unpredictable results for the future development of enterprises. Compared with the traditional intensive management mode, the rapid and iterative innovation mode brought by the availability and readability of digital infrastructure is obviously more efficient and dynamic, thus digital innovation, inevitably becoming incredibly unpredictable in all likelihood [61]. The impact of big data capability on supply chain performance and even competitiveness, surprisingly enough, is amplified in the rapidly developing digital economy market [62]. It is certain that experts in the field of information management confirm that enhanced environmental dynamics is beneficial to the correlation of digital technology with performance [63].

It is clear, then, that a high-munificent environment facilitates firms to access the resources needed to develop and transform business models at low transaction costs, thus reducing the difficulty and cost of innovation [28]. Back to the research topic, the rapid development of digital technology has improved the storage and transmission speed of information and knowledge, together with reducing the cost of communication and

search. Enterprises, driven by digital transformation, are forced to identify, acquire, and reallocate resources from the environment for existence, thus creating a dependence on the environment [64]. Due to the lack of an inclusive digital environment, firms devoid the necessary resource support when launching innovation strategies, in all likelihood, thus demand for external resources increases and the industry competition becomes more serious. This may act as a reverse incentive for enterprises to continuously scan and update the digital environment in the opposite perspective [65]. The complexity of the environment (dynamics and munificence) includes rapid changes in consumer demand, industry competition, policy support or crackdown, which would bring more unpredictable results for the future development of enterprises.

## 3. Methodology

Given that the net impact of independent variables on results in traditional regression statistical methods is impossible to explain the content structure and logical mechanism behind digital innovation perfectly, we recognize the necessity for exploring a combination of multiple factors and their configuration paths from an all-round perspective. We employ fuzzy set qualitative comparative analysis (fsQCA) to examine the interaction of different causes and results. Furthermore, the non-conflicting equivalent relationship between different combinations of explanatory variables and results clarified the factor combinations that generate high and non-high result. As mentioned above, we combine an exploratory induction, as well as a deductive method grounded in the TOE theoretical framework, to conduct fsQCA analysis. In the process of data collection, the sample size should fall large samples and conventional case studies in between [66], as well as the applicability of the scale in the current situation should be tested by means of a pre-test, which is regarded as an effective means to test the usability of questionnaires to ensure the quality of large-scale surveys.

### 3.1. Data Collection

How to promote the enterprise digital innovation practice is an urgent problem to be solved in the current stage of digital transformation and upgrading of manufacturing enterprises. Given this fact, the paper selects private high-tech enterprises in the Yangtze River Delta region with good development of digital transformation and upgrading of China's manufacturing industry to discuss the solution mechanism of the above problems. The following sample selection conditions: the enterprise has been established for more than 3 years and has entered the stage of deep application from the initial transformation of digitalization; the primary data of enterprises are significantly available and abundant. Different types of manufacturing enterprises are covered to highlight the differences between samples.

We employed sample sources of alumni union and EMBA students from Yangtze River Delta, and of distribute questionnaires through cooperation with the China Digital Economic Industry Alliance (DEIA). Considering the negative impact of the common method bias [67], the same top manager was investigated at different time points, and two questionnaires were designed to be filled in by the CEO and one TMT member, respectively, for comparison and reference. In answering the questionnaire, respondents were requested to assess their firm's level of digital transformation and innovation performance over the past several years and complete the questionnaires in detail with real perception. On the basis of absolute confidentiality, respondents were asked to guarantee no false answers to minimize social desirability bias.

To investigate whether these items have semantic differences that affect respondents' understanding of the content, we conducted a pre-test from January to mid-April 2022, successfully collected 254 paired questionnaires, which passed the tests of CITC analysis and internal consistency analysis, indicating the applicability of each item in the current situation. However, due to the lack of objective data of several companies and several invalid questionnaires with inconsistent answers, as well as some respondents' inquiries

about the meaning of the items during the recycling process, we modified the expression of individual items to improve the clear understanding in the Chinese context. In the end, 176 companies that submitted valid data were officially investigated and date were collected from August to mid-September 2022. Comparing the responses filled out by the same respondents in the first round and keeping a consistent questionnaire, we obtained a final 141 valid questionnaires by excluding incomplete surveys and those answered by firms whose CEO was not the key informant.

### 3.2. Measurement of Antecedent Condition

Technological affordance. It has been stated explicitly that IT functions, and the way in which organizations and their social structures use these functions, can be defined as affordance [68]. The measurement of IT affordance by Ref. [68] has been widely applied and proven [48]. Ref. [69] measured digital affordance in terms of generativity and disintermediation. In view of Refs. [48,68] repeated validation in the Chinese context, we conducted a two-dimensional test of technological affordance for accumulation and variation. The former includes four items related to homogeneity, including "We are able to conduct data analysis for various businesses such as R&D, design, manufacturing, and product services," while the latter includes three items related to reprogramming, including "We are fully advancing digital design, production, and management."

Technological complexity. Measurements of technological complexity in prior studies are widely based on patent data for analysis [50]. While ref. [70] aimed to measure electric/electronic (E/E)-technologies, developing 12 items, including components, control units, event chains, multi-functionalization, subsystem correlation, etc., ref. [71]) designed a scale of three items about the IS complexity in terms of operation, technical assistance, and skill difficulty. Considering that ref. [72]'s measurement in terms of the number of design schemes, technical novelty, and span of new knowledge is more widely applicable, items involving parts (components) are removed and six items including "Technological innovation requires a high degree of involvement and participation of users and suppliers" are formed.

Organizational agility. Different scholars adopt different dimensions for measuring organizational agility, such as customer responsiveness [73], partner agility [74], operational flexibility [75]. As well as refs. [76,77]'s recommendation on the company's perception and response to changes in marketplaces. Given that more studies tend to synthesize the dimensions mentioned above to construct the scale [52], we measured organizational agility from three aspects: customers, partners, and operations. Sample items, such as "We always pay attention to the behavior of competitors", and "We can quickly develop and implement the competitive strategy in response to the behavior of competitors."

Organizational unlearning. Given that unlearning is a learning/relearning process in which new routines and beliefs are established, except eliminating old cognitive patterns [78], ref. [79] measured organizational unlearning from this perspective. Ref. [80] learned from three of them, including the improvement of product development, information sharing, and decision-making process, which were certified in the Chinese context. We refer to the assessment items by ref. [79] that contain five items as "We can continuously improve the decision-making process of the project team."

Environmental dynamics. One of the methods for measuring environmental dynamics in literature is to determine the fluctuations of enterprise activities affected by external environmental changes, drawing on objective indicators such as sales revenue and EBIT [81]. Considering the difficulty of collecting accurate public data of SMEs and the small sample size, we refer to the dynamic subjective measurement method of the environment of ref. [82] to measure the degree of technological change and innovation, the change of customer demand and preference, and the degree of industry competition. That is, a questionnaire that contained nine perceived questions as "Clients tend to look for new products all the time.

Environmental munificence. Ref. [83] measured environmental munificence in terms of financial support, profit opportunities, external threats, resource availability, etc. Ref. [84] developed five assessments to measure environmental munificence through government policies, infrastructure construction, stakeholder relations, sociocultural, and social concerns. This scale has been verified in the Chinese context, which further supports the situational feasibility of the scale [85]. Sample items, such as "Bankers and other investors go out of their way to help new firms get started".

### 3.3. Assessment of Dependent Variable

How to measure enterprise digital innovation has always been the focus of scholars and entrepreneurs [86]. In line with the previous literature on innovation strategy, the number and growth of patents of digital products can represent the innovation activities of enterprises [56], but there are also problems of difficulty in obtaining and single evaluation, because digital innovation is not only the development of new products through the interconnection and combination of digital technologies and products or services, it also includes process transformation, organizational transformation, and business model innovation [5,6]. Therefore, we refer to the research results of ref. [86] to measure a firm's digital innovation from the aspects of exploiting digital opportunities, inventing digital products/services, improving business processes, and improving organizational operational efficiency. The measures of the constructs were adapted from existing scales from previous studies. The 5-point Likert scale, ranging from 1 ("Strongly disagree") to 5 ("Strongly agree"), was used to evaluate the multi-item constructs.

### 3.4. Data Analysis

The first step of our data analysis was to use the reliability coefficients (C$\alpha$), composite reliabilities (CR), and average variance extracted (AVE) to assess reliability estimates of the key variables, as shown in the Table 1, The C$\alpha$ and CR values were above the threshold of 0.70, and the AVE values were above the recommended threshold of 0.50, indicating that each index had good reliability. In addition, factor load and the square root of the standard deviation can be respectively used to measure the discriminative validity, convergent validity and found at the $p < 0.01$ significant level, the factor loading were greater than 0.7, the square root of AVE is greater than its various factors and the correlation coefficient of other factors, shows that the measurement model has good convergent validity and discriminative validity.

**Table 1.** Results of reliability and validity.

| Results and Conditions | | C$\alpha$ | CR | AVE | Minimum Factor Load |
|---|---|---|---|---|---|
| **Technology** | Affordance (TA) | 0.781 | 0.875 | 0.603 | 0.704 |
| | Complexity (TC) | 0.802 | 0.873 | 0.623 | 0.756 |
| **Organization** | Agility (OA) | 0.796 | 0.831 | 0.614 | 0.732 |
| | Unlearning (OU) | 0.733 | 0.866 | 0.598 | 0.812 |
| **Environment** | Dynamics (ED) | 0.821 | 0.889 | 0.607 | 0.789 |
| | Munificence (EM) | 0.807 | 0.867 | 0.637 | 0.745 |
| **Digital innovation (DI)** | | 0.719 | 0.841 | 0.594 | 0.751 |

### 3.5. Data Calibration and Necessity Analysis

Prior to constructing the dataset, a critical step in fsQCA analysis is to calibrate the original data to obtain fuzzy membership scores, so as to fully account for the category and degree differences among cases [87]. Present studies advocate the direct calibration method to transform the original data into the distribution of full membership, cross-over point, and full non-membership threshold by logical functions [88]. Therefore, we calculated the mean of the seven constructs, together with setting the calibration critical values corresponding to 0.95, 0.5, and 0.05 degrees of membership, respectively. As mentioned above, considering the 7-point Likert scale adopted and the suggestions of Ref. [89], we

consider the mean value as the critical value of 0.5 membership degree, along with the minimum and maximum values of the mean value corresponding to full-non-membership and full membership, thus the correction results of the data shown in Table 2.

**Table 2.** Calibration values at extreme values and crossover point of each condition (*n* = 141).

| Statistic | DI | TA | TC | OA | OU | ED | EM |
|---|---|---|---|---|---|---|---|
| Full membership | 7 | 7 | 7 | 7 | 7 | 7 | 7 |
| Cross-over Point | 4.041 | 5.175 | 4.821 | 5.032 | 4.987 | 4.548 | 4.679 |
| Full- Non-membership | 1.33 | 1 | 1.43 | 2 | 1 | 1.83 | 1.33 |

We need to investigate whether the degree of consistency of cases with common condition configuration belonging to the same result is higher than the acceptable empirical standard of 0.9, which is similar to the expression of significance in regression analysis [90]. Causal complexity causes overlapping of results among configurations, and the extent to which ensemble relationships via consistency tests explain results is analyzed by reporting coverage (strength in correlation analysis) [88]. Table 3 exhibited the results of necessary conditions for the firm's digital innovation, in which the consistency level of all conditions is no higher than 0.9. in other words, a single condition cannot constitute the necessary condition to affect the results, which means that the complexity of enterprise digital innovation is jointly affected by multiple factors.

**Table 3.** Analysis results of necessary conditions.

| Condition Variable | High Digital Innovation | | Non-High Digital Innovation | |
|---|---|---|---|---|
| | Consistency | Coverage | Consistency | Coverage |
| TA | 0.735 | 0.687 | 0.681 | 0.663 |
| ~TA | 0.324 | 0.371 | 0.524 | 0.549 |
| TC | 0.819 | 0.773 | 0.752 | 0.601 |
| ~TC | 0.387 | 0.368 | 0.598 | 0.633 |
| OA | 0.891 | 0.781 | 0.651 | 0.698 |
| ~OA | 0.415 | 0.369 | 0.715 | 0.797 |
| OU | 0.768 | 0.806 | 0.729 | 0.762 |
| ~OU | 0.361 | 0.378 | 0.684 | 0.554 |
| ED | 0.685 | 0.602 | 0.814 | 0.788 |
| ~ED | 0.511 | 0.453 | 0.471 | 0.372 |
| EM | 0.714 | 0.735 | 0.705 | 0.687 |
| ~EM | 0.397 | 0.381 | 0.595 | 0.603 |

## 4. Result

Combined with the number and distribution characteristics of this case, the minimum acceptable threshold of 0.8 proposed by experts was taken as the standard [88]. Similarly, in order to avoid the phenomenon of contradictory configuration, PRI consistency is used to effectively reflect the degree of high digital innovation of the truth table row by taking the lowest acceptable standard 0.7 as the threshold [90]. The condition that appears in both intermediate solution and personified solution is set as the core condition, while the condition that only appears in the intermediate solution is the edge condition [88]. Thus, fsQCA3.0 software is applied to obtain different combination effects of different configurations on high digital innovation and non-high digital innovation. As shown in Table 4, we identified four combinations of explanatory variables with high digital innovation and one with non-high digital innovation, thus differences of them providing a holistic perspective for us to analyze the portfolio effect model of SMMEs with high digital innovation. Firstly, the diversified path to generate high digital innovation includes five condition configurations, namely H1a, H1b, H2, H3, and H4. Among them, the second-order equivalent configuration formed by H1a and H1b means their core conditions are

consistent [91]. The consistency index (total consistency = 0.811) indicated that the five configuration groups constituted sufficient conditions for digital innovation. Together with the coverage (total coverage = 0.671) indicator, it shows the substantial explanatory power of each configuration for high digital innovation. Secondly, we find that there is only one path that affects non-high digital innovation in enterprises, among which the overall consistency is 0.853 and the overall coverage is 0.103, indicating that the explanatory power of this configuration for sample cases is more than 10%, which meets the criteria of adoption analysis.

**Table 4.** Configuration of SMMEs with high and non-high digital innovation performance in fsQCA.

| Condition Variable | High Digital Innovation | | | | | Non-High Digital Innovation |
|---|---|---|---|---|---|---|
| | H1a | H1b | H2 | H3 | H4 | H5 |
| TA | ● | ● | ⊗ | ● | ● | ⊗ |
| TC | ⊗ | ⊗ | ⊗ | ● | ● | |
| OA | ⊗ | ⊗ | ● | ● | ● | ⊗ |
| OU | ⊗ | ● | ● | | ● | ● |
| ED | | ● | ● | ⊗ | ● | ● |
| EM | ● | ⊗ | ⊗ | ⊗ | | ● |
| Consistency | 0.903 | 0.914 | 0.876 | 0.927 | 0.884 | 0.811 |
| Raw coverage | 0.156 | 0.187 | 0.165 | 0.139 | 0.087 | 0.103 |
| Unique coverage | 0.045 | 0.081 | 0.077 | 0.105 | 0.062 | 0.052 |
| Solution consistency | | | 0.914 | | | 0.811 |
| Solution coverage | | | 0.316 | | | 0.103 |

Notes: "●", "⊗" denotes the existence and absence of core conditions respectively; "" indicates that the condition may or may not occur.

### 4.1. Robustness

The QCA method adopts the robustness test to investigate the sensitivity and randomness of the results, so as to avoid different results that may be caused by the differences in the inclusion conditions in the study [43]. By increasing the PRI consistency threshold to 0.8, the configuration is recalculated to obtain Table 5, which shows that the consistency of the overall solution is improved to 0.921, and the coverage is reduced to 0.217. The three configurations of the new model are completely consistent with those of the original model, and together with clear subset relations exist in configurations H3 and H5. This is consistent with the theoretical logic that the new configuration is a subset of the original one because it is difficult to maximize the simplified configuration after increasing the consistency threshold.

**Table 5.** Robustness test of configuration.

| Condition Variable | H1a | H1b | H2 | H3 | H4 | H5 |
|---|---|---|---|---|---|---|
| TA | ● | ● | ⊗ | ● | ● | ⊗ |
| TC | ⊗ | ⊗ | ⊗ | ⊗ | ● | |
| OA | ⊗ | ⊗ | ● | ● | ● | ⊗ |
| OU | ⊗ | ● | ● | ⊗ | ● | ⊗ |
| ED | | ● | ● | ⊗ | ● | ● |
| EM | ● | ⊗ | ⊗ | ⊗ | | ⊗ |
| Solution consistency | | | 0.921 | | | 0.801 |
| Solution coverage | | | 0.217 | | | 0.096 |

Notes: "●", "⊗" denotes the existence and absence of core conditions respectively; "" indicates that the condition may or may not occur.

### 4.2. TOE Framework That Generates High Digital Innovation

Technology-environment oriented. Configuration H1a is the technology affordance driven type supported by munificent environment (Consistency = 0.903, Raw coverage = 0.156). Regardless of whether the business environment is volatile and complex, provided that

organizations fully grasp and utilize the advantages of digital technology affordance in the context of simultaneously underperforming complexity, agility, and unlearning, a higher level of digital innovation is highly likely to be activated and supported by environmental inclusion. A plausible possibility is that the homogeneous processing and reprogrammability process of digital technology affordance provides enterprises with a variety of innovation paths based on technology. Along with their strong dependence on the external environment, the increase of resource acquisition opportunities and the reduction of acquisition difficulty are significantly conducive to the pursuit of high innovation. This path indicates that most of the SMEs that realize digital innovation have a high degree of digital technology readiness and maturity before embracing innovation.

Organization-technology oriented. Configuration H3 is an agile driven type with technology support (Consistency = 0.927, Raw coverage = 0.139). Regardless of whether the unlearning ability of the organization is prominent or not, provided that digital technology contains two important characteristics and maintains the agility of the organization, high digital innovation could still be created even if the external environment is not inclusive and dynamic. A possible reason is that the dominant characteristics of the complex interaction of digital technology itself to activate the organizational learning, resource adjustment, the dynamics of knowledge sharing in the process to establish a set of resources, demand rapid response and the reconstruction process model, the dynamic organization mode, and technical support have become an important weapon guaranteeing the realization of digital innovation for a long time. Therefore, this path significantly enables enterprise innovation for many agile organizations in the reconfigured digital age, together with a moderate level of digital technology.

Organization oriented-environment. Configuration H2 is the unlearning driven model supported by a dynamic environment of an agile organization (Consistency = 0.876, Raw coverage = 0.165). This states that firms may have insufficient understanding and utilization of the availability and complexity of digital technology, as well as the lack of resources needed for the development of opportunities. Instead, the strong adaptability of the organization to the changeable background, timely discard of the old management paradigm and decision-making strategy, so as to promote the sustainable innovation of the enterprise through the reallocation of resources. The innovation practice scheme of technology-deficient enterprises has become a prominent representative of this path, that is, technology is not the necessary core condition for enterprises to achieve digital innovation, considering that the subject with technological disadvantages should primarily focus on the symbiosis of organization and environment.

Total factor driven. Configuration H1b describes the all-factor drive path of affordance-unlearning-dynamic (Consistency = 914, Raw coverage = 0.187). Given such complex characteristics of digital technology as openness, dynamics, and extensibility, no matter the physical components included in the technology or the cognition and usage of the technology, thus less important the complexity. Uncertain environment characteristics, further, indirectly strengthened the enterprise to the difficulty of resource acquisition, what is not more difficult is that the core ability of enterprises to respond to the external environment is weakened, but under the interactive influence of availability and dynamics, unlearning plays a driving role in the continuous iterative process of innovation. It is obvious that the synergistic effect of the above factors leads to the successful path of innovation. Similarly, configuration H4 is based on various conditions to jointly promote enterprise digital innovation via mutual linkage and adaptation. In this case, we could ignore the tolerance degree of the environment, together with the support of available and complex technology, the unlearning agile organization constructed has stronger vitality in the face of dynamic environment, which provides practical reference for enterprises to maintain the core position of affordance, agility, and unlearning.

*4.3. Toe Framework for Generating Non-High Digital Innovations*

Configuration H5 draws the antecedent configuration of non-high digital innovation (Consistency = 0.811, Raw coverage = 0.103). The main reason for non-high innovation is to attach much importance to unlearning and the utilization of the changeable or munificent environment, while ignoring the value of technical affordance and the construction of an agile organization. Enterprises with weak situational capability and dynamic capability of digital technology application certainly will make a strategic adjustment from both technical and organizational aspects.

## 5. Discussion

Given the speed with which digital technologies are disrupting industries globally, it is inevitable that companies will innovate to stay competitive as they embrace digital transformation [3,5,6,34]. Unfortunately, we lack a comprehensive understanding of how organizations strategically apply, leverage, and integrate digitalization for digital innovation. The purpose of this article is to expand our clarity and understanding of the key elements involved in the generation of digital innovation processes and outcomes. Previous research on digital technologies and strategic management has examined the role of physical elements and organizational capabilities in activating digital innovation in a decentralized manner [21]. We propose a new conceptual framework by integrating the realizable conditions of digital innovation into the TOE framework. Based on the data collected from 141 SMMEs in China, we test different antecedent configuration research models to explain digital innovation. Consistent with previous studies, technology, organization, and environment have been proved to be the key antecedents supporting the successful practice of digital innovation [16,47,73]. Our empirical research results further support that matching different antecedents leads to different innovation paths, which conforms to the configuration analysis that the paths to produce unified results are diverse and equivalent [87]. Together with digital innovation research, the focus has shifted to single-line causality and linear relationships between variables.

We could explicitly discover that the core conditions and auxiliary conditions presented in the path H1 to H5 are different and interconnected but would lead to the same results in all likelihood. Specifically, configuration H1a (technology-environment oriented) emphasizes the unsubstitutability of affordance to facilitate organizations to quickly complete communication and collaboration under the same standard, together with the realization of reprogrammable innovation through different combinations of digital technologies, which is widely applicable to digital entrepreneurial companies that have mastered and applied digital technologies. Configuration H2 (organization oriented-environment) focuses on the company's dynamic environment response ability and unlearning ability. The complex digital environment forces the company to abandon the old management paradigm and reconfigure resources to promote sustainable innovation, which is the key method for the technologically disadvantaged company to choose the digital strategic transformation. Configuration H3 (organization-technology oriented) describes the dynamic adjustment process of activated resources depending on technical features. Digital agile organizations will become an important weapon for general companies to achieve continuous digital innovation. Configuration H1b and H4 (total factor driven) confirm that the configuration linkage of each condition under TOE framework jointly promotes digital innovation, focuses on the technical accessibility and organizational ability as the core conditions, and further cultivates the vitality of digital agile organization by using resources given by external environment. H5's non-high digital innovation path, as mentioned above, further emphasizes the value of technological affordance and agility. As the digital innovation realization path encounters obstacles, it is requested to shift the focus from the transition of the environment to the strategic adjustment of situational application with technology and dynamic capability.

## 5.1. Theoretical Implications

Digital innovation is undoubtedly an intricate iterative process [9]. Small and medium-sized manufacturing enterprises are subject to the shortage of resources and technologies. Compared with large enterprises, the launch, development, and application of digital innovation are more complex and arduous [18]. This study provides some theoretical contributions for reference in comparison to the current research: on the one hand, it reveals the supporting forces and driving factors of high digital innovation in small and medium-sized manufacturing enterprises, and more importantly, it explores the joint effects and interaction forces of six conditional variables on digital innovation. Previous studies focused on technology and organization as a single condition, together with environmental uncertainty characteristics as moderating variables, to explore the impact on innovation or digitalization via correlation and regression analysis [21]. We more comprehensively and systematically constructed the digital driven innovation mechanism of the interaction results of three elements, that is, the fsQCA method was adopted to provide a holistic perspective for understanding and explaining the influencing factors and causal complexity of digital innovation [92]. The TOE theoretical framework, on the other hand, is introduced into the research of enterprise digital innovation, which not only enriches the perspective of digital innovation research, but also broadens the application scope of the TOE model, that is to say, it breaks through the traditional application scenario analysis framework based on adopting innovative technologies. This is a response to existing research calling for practice in the areas of government policy, business model innovation, performance, and digital transformation [40–42], further exploring the specific connotation of the TOE framework in the Chinese context.

## 5.2. Practical Implications

We propose the following countermeasures for the construction and application of firm digital innovation: First, SMMEs should attach importance to and utilize the linkage integration of technology, organization, and environment conditions, but measures should be adapted to local conditions. Second, focusing on exploiting and releasing digital technology affordance, the company uses digital technology to enhance or to replace the traditional production technology, for upgrading product innovation in response to user requirements. Thirdly, there is a need to build an agile organization with customers, partners, and operations as the core to cope with the changing digital age. Fourthly, the company takes the initiative to forget the outdated knowledge, such as cognitive thinking, management mode, technology and working methods that are inconsistent with digital transformation and innovation development. In particular, the company no longer relies on the successful experience, path, and paradigm obtained in the past, but establishes a new learning organization according to the innovation requirements of the digital era. Finally, identifying and analyzing the dynamic changes of market, technology, customer demand, and competitors has become an indispensable ability for a firm's development; it is necessary to extract the available and transformed resources, regularly evaluate the complexity and tolerance of the environment, together with adjusting the organizational strategy and learning ability.

## 6. Limitations and Future Research

Although the TOE analysis framework has covered a variety of influencing factors, we place emphasis on the characteristics and performance of technology–organization–environment, but do not involve the specific quantifiable content including firm technology infrastructure, organizational digital strategy, government policy support, and so on. Given that setting overmuch method restrictions on the method of condition variables is not recommended, future research could also consider more potential success factors from different theoretical perspectives and dimensions to build a more comprehensive and effective analysis model for firm digital innovation. Furthermore, we only explore the static relation with antecedent configuration and digital innovation, while the dynamic

iterative process of digital innovation is also worthy of attention. Future research would use the time-series QCA analysis method to explore the complex impact of the evolution of different conditions configuration on digital innovation. Finally, we admit that the data features of the questionnaire have structural advantages, but the subjective data have the disadvantage of an insufficient in-depth case phenomenon. In the future, we will consider using grounded theory and the open enterprise case database to collect and analyze a variety of data, so as to strengthen the objectivity and representation of data.

**Author Contributions:** Conceptualization, F.L.; Software, F.L.; Validation, F.L.; Investigation, F.L.; Resources, W.Z.; Data curation, W.Z.; Writing—original draft, F.L.; Writing—review & editing, J.L.; Supervision, J.L. All authors have read and agreed to the published version of the manuscript.

**Funding:** Innovation Capability Support Program of Shaanxi (Program No. 2022KRM104).

**Conflicts of Interest:** The authors declare no conflict of interest.

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
