# Peer review of "Mining Braces of Innovation Linking to Digital Transformation Grounded in TOE Framework"

_sustainability, doi:10.3390/su15010301_

Round 1

Reviewer 1 Report

the topic of the paper is interesting

unfortunately, the topic of the paper seems to have no relation whatsoever to sustainability

although the abstract has some strange english sentences, the summary of the paper is very clear

the introduction has very long sentences and paragraphs, making the text difficult to read

there are some small language errors, such as "organizationalunlearning" and "thatreasonably" and "unlearningabilityconducestomake"

the research topic, research objectives, research questions and structure of the paper are presented in the introduction

however, the introduction should also include a paragraph about the research methodology

the introductino should also clarify the difference between digital projects, digital transformation and digital innovation

section 2 (Literature review and theoretical framework) presents the "research model" based on the TOE framework but, since the indicators were chosen by the authors, in fact the model should be presented more like a proposal

the methodology section 3 finally explains the paper is based on a survey with 141 answers, but it's not clear why two rounds were performed

although the paper tries to explain and justify the choices, not only the indicators were chosen by the authors, but also the questions to measure the indicators were chosen by the authors! that means the model is basically a proposal (that could be evaluated by this survey) but, on the other hand, any conclusions from the survey regarding digital innovation should be taken with care!

this ad-hoc approach to the research model can be exemplified by this sentence "The measures of the constructs were adapted from existing scales from previous studies" in section 3.3 that explains and justifies almost nothing

i am not an expert, so i will just assume the statistical analysis is well done (other reviewers will certainly comment on this)

the results are then presented in section 4. the "five condition configurations" (explained in detail) are very interesting!

the robustness section should be presented *before* the "five condition configurations"

section 5 (discussion) describes the theoretical and practical implications, but the "Limitations and future research" should be moved to a new Conclusion section (that would start with the main contributions)

Author Response

请参阅附件。

Reviewer 2 Report

There are some grammar and Orto-typographic errors, please check….(all in red)

Introduction,

The antecedents, the reasons why it is interesting doing this paper are Ok and citations are good.

However, in the last part of the introduction I perceive a lack of structure.

Authors mix research questions and objectives with practical implications, and they should be clearly distinguished.

So, I recommend at the end of the introduction, once that evidences and citations are provided to include first research questions (considering the previous facts, what are the main research questions that we want to answer in this paper?), second, I suggest authors include objectives (of course aligned to research questions), and third, I suggest authors anticipate a little main theoretical and practical implications derived from the paper.

And just at the end of the introduction, I miss a last paragraph explaining the structure of the paper. Something that starts for example: After this introduction, in part 2 the literature review and the theoretical framework are described, and then….and last conclusions…

There is a need to include citations for some statements provided (indicated as comments in the text).

In methodology, in data collection, some citations are also required to support the way data have been collected and the methodology used (can authors mention some other publications applying same methodology?)

In the part: 3.2. Measurement of Antecedent condition, when the different ways of measuring variables is explained, I suggest that apart from explaining the inspiration for the measure, to cite some previous research that has taken into consideration each of the measures.

5. Discussion is poor, there is not discussion. What this paper agrees with other similar ones? What is different and original and why? From results what can be re-explained connected with author’s own experiences and knowledge in this field?

Please extent the discussion facing what it has already been concluded in some other analysis and enriching the results with your own knowledge and experience from a critical perspective.

Round 2

Reviewer 1 Report

although the relationship between digital transformation and sustainability is very thin (to say the least) the authors have now included a small text in the introduction to justify why this paper makes sense in a journal about sustainability

the most obvious language errors in the text have now been removed

the authors introduced a few lines about the research methodology, but a better explanation about the research methodology should be included in the section 3 just before section 3.1

the authors have now explained the "research model" a little bit better, although it still seems arbitrary -- having said that, it's clear now the "research model" is just an hypothesis that will be evaluated later in the paper

the need for two rounds in the survey is now explained

the indicators and measures are now better explained

the robustness section has been moved

Reviewer 2 Report

I think the paper has increased quality in this second revision. Some aspects related to the introduction, references and objectives have been improved clearly, however there are others that still require attention from authors:

On the one hand, I recommend including in the abstract most outstanding practical implications, but not so general, be more concrete, with at least a couple of them.

On the other hand, discussion must be improved. It is very general now, and different aspects must be discussed in deep. I suggest discuss one by one the different results obtained for each hypothesis.
